# Dual Modulator of ASIC Channels and GABA_A_ Receptors from Thyme Alters Fear-Related Hippocampal Activity

**DOI:** 10.3390/ijms241713148

**Published:** 2023-08-24

**Authors:** Aleksandr P. Kalinovskii, Anton P. Pushkarev, Anastasia D. Mikhailenko, Denis S. Kudryavtsev, Olga A. Belozerova, Vladimir I. Shmygarev, Oleg N. Yatskin, Yuliya V. Korolkova, Sergey A. Kozlov, Dmitry I. Osmakov, Alexander Popov, Yaroslav A. Andreev

**Affiliations:** 1Shemyakin-Ovchinnikov Institute of Bioorganic Chemistry, Russian Academy of Sciences, ul. Miklukho-Maklaya 16/10, 117997 Moscow, Russiakudryavtsev@ibch.ru (D.S.K.); o.belozyorova@gmail.com (O.A.B.); serg@ibch.ru (S.A.K.); popov@neuro.nnov.ru (A.P.); aya@ibch.ru (Y.A.A.); 2Moscow State Academy of Veterinary Medicine and Biotechnology—MVA named after K.I. Skryabin, ul. Akademika Skryabina, 23, 109472 Moscow, Russia; 3Institute of Molecular Medicine, Sechenov First Moscow State Medical University, Trubetskaya Str. 8, bld. 2, 119991 Moscow, Russia

**Keywords:** acid-sensing ion channels, γ-aminobutyric acid receptors, open field test, passive avoidance test, θ rhythm, anxiety, in vivo electrophysiology

## Abstract

Acid-sensing ion channels (ASICs) are proton-gated ion channels that mediate nociception in the peripheral nervous system and contribute to fear and learning in the central nervous system. Sevanol was reported previously as a naturally-occurring ASIC inhibitor from thyme with favorable analgesic and anti-inflammatory activity. Using electrophysiological methods, we found that in the high micromolar range, the compound effectively inhibited homomeric ASIC1a and, in sub- and low-micromolar ranges, positively modulated the currents of α1β2γ2 GABA_A_ receptors. Next, we tested the compound in anxiety-related behavior models using a targeted delivery into the hippocampus with parallel electroencephalographic measurements. In the open field, 6 µM sevanol reduced both locomotor and θ-rhythmic activity similar to GABA, suggesting a primary action on the GABAergic system. At 300 μM, sevanol markedly suppressed passive avoidance behavior, implying alterations in conditioned fear memory. The observed effects could be linked to distinct mechanisms involving GABA_A_R and ASIC1a. These results elaborate the preclinical profile of sevanol as a candidate for drug development and support the role of ASIC channels in fear-related functions of the hippocampus.

## 1. Introduction

Anxiety disorders are mental conditions of great public health significance that feature excessive and enduring fear, anxiety, or avoidance of perceived threats, and possible panic attacks and can cause social maladaptation and severe disabilities. The management of anxiety disorders can be based on cognitive–behavioral therapy and/or pharmacological intervention. The medical drugs in use include antidepressants (selective serotonin reuptake inhibitors and serotonin–noradrenaline reuptake inhibitors) and benzodiazepines [1]. Benzodiazepines enhance inhibitory neurotransmission mediated by γ-aminobutyric acid (GABA) by acting as positive allosteric modulators of ionotropic GABA receptors (GABA_A_R). GABA_A_R are heteromeric pentamers that, in vertebrate nervous systems, can be composed of twenty isoforms (α1-6, β1-4, γ1-3, δ, ε, π, ρ1-3). Popular in the past, benzodiazepines are now being replaced with drugs with fewer side effects and lower risks of abuse [2].

Acid-sensing ion channels (ASICs) are proton-gated ion channels from the degenerin/epithelial sodium channel (ENaC/DEG) superfamily that are abundant in the nervous system. Six subunit isoforms of ASICs (ASIC1a-b, 2, 3, 4) form homo- and heterotrimeric channels that differ in sensitivity to protons and tissue-specific expression [3]. In the periphery, ASIC1a, ASIC2, and, most importantly, ASIC3 contribute to the transduction of mechanical, chemical, and nociceptive signals via sensory neurons and other types of cells. For instance, ASIC3 is required for the development of acute and chronic pain following intramuscular and intraperitoneal injectionsof acid [4]. In the central nervous system (CNS), ASIC channels participate in the cognitive and emotional functions of the brain. ASIC1 channels are abundant in the hippocampus and cerebellar cortex, where they contribute to synaptic plasticity, learning, and memory [5,6]. ASIC1a was detected in the medial amygdala and dorsal periaqueductal gray, which are involved in the generation and expression of innate fear. Genetic deletion of ASIC1a altered neuronal activity in these structures, which was accompanied by reduced unconditioned fear behavior in an open field test, reduced acoustic startle, and inhibited fear response to the predator’s odor [7]. Restoring ASIC1a in the basolateral amygdala of ASIC1a-null mice rescued context-dependent fear memory but not the freezing deficit during training or the unconditioned fear response to predator odor [8]. ASIC1a in the amygdala was associated with fear behavior following inhalation of a high content of carbon dioxide and sensing the subsequent acidosis, which may underlie anxiety and panic disorders [9]. Certain single-nucleotide polymorphisms in the ASIC1 gene are highly represented in individuals with panic disorders and increased reactivity and volume of the amygdala [10,11]. Also, it has beenshown that ASIC1a in the ventral hippocampus is required for the regulation of fear extinction via hippocampal–prefrontal interaction [12].

The distinct role of ASIC1a in anxiety-related processes has led researchers to attempt employing ASIC inhibitors as anxiolytics. Intracerebroventricular (i.c.v.) infusion of the *Psalmapoeus cambridgei* tarantula venom, which contains a potent inhibitor of ASIC1a, psalmotoxin-1 (PcTx-1), reduced freezing in mice evoked by a predator’s odor but did not affect freezing in ASIC1a-null mice [7]. Amiloride, a non-selective blocker of ENaC/DEG channels, A-317567, a blocker selective to ASICs over other ENaC/DEG channels, and PcTx-1, exhibited differential anxiolytic-like activities that were dependent on the studied model [13].

From a pharmacological point of view, ligands of natural origin are more attractive for the development of drugs with a minimum set of side effects [14]. One is sevanol, a naturally-occurring lignan isolated as a major component of the acidic extract from thyme *Thymus armeniacus*. Sevanol reversibly inhibits both transient and sustained currents of rat ASIC3 and the current of rat ASIC1a. At doses of 1–10 mg/kg i.v., sevanol demonstrated a pronounced analgesic effect in an acetic acid-induced writhing test and reversed thermal hyperalgesia induced by Complete Freund’s Adjuvant (CFA). Doses of 0.1–1 mg/kg p.o. also significantly relieved paw inflammation induced by CFA [15,16]. Favorable efficacy in vivo indicates the potential of sevanol as an analgesic therapeutic; however, its activity against anxiety-related conditions hasnot been investigated before. In this study, using an optically pure sample of sevanol produced by chemical synthesis, we measured its activity on homomeric ASIC1a in a model of human dopaminergic neurons and on GABA_A_R in a heterologous expression system. For the first time, we report that sevanol positively modulates α1β2γ2 GAB_A_A receptors. Next, we studied sevanol action in behavioral models relevant to anxiety using the delivery of the drug via cannulas into the hippocampus and simultaneous electroencephalographic recordings. These results provide the first evidence of the possibility of employing sevanol for comorbid anxiety states and investigate the role of ASIC channels in passive avoidance learning.

## 2. Results

### 2.1. Preparation and Testing of Optically Pure Sevanol

An efficient scheme for the preparative production of sevanol via chemical synthesis has been previously developed [16,17]. The yielding product was a mixture of diastereomers, sevanol, and isosevanol, in a ratio of 3:1, that differed in the configuration of the C1-C1’ bonds in the dihydronaphthalene core of the molecule (Figure 1A). In order to assess the impact of isosevanol on the biological activity of sevanol alone, diastereomers were separated by a simple and effective scheme using a C_18_ stationary reversed-phase column for high-performance liquid chromatography (HPLC) in the presence of 0.1% trifluoroacetic acid (TFA). The use of an isocratic elution by 10% acetonitrile was more effective in separating the two nearest diastereomers than a prolonged gradient elution (0.5% solvent B/min) (Figure 1B,C). Eventually, HPLC, in the isocratic mode, allowed the load capacity to be increased 6-fold without lowering the quality of the separation.

The inhibitory activity of optically pure sevanol to rat ASIC1a (rASIC1a) was evaluated with two-electrode voltage-clamp (TEVC) recordings on the channels heterologously expressed in oocytes of *Xenopus laevis*. rASIC1a currents were elicited by a rapid application of the activating solution with pH 6.0 in a bath conditioning at pH 7.4. Sevanol dose–dependently inhibited pH 6.0-induced currents, achieving the complete block at 600 µM (Figure 2A,B). The fitting of the dose–response data with Hill’s equation gave the concentration of half-maximal inhibition (IC_50_) equal to 198 ± 19 µM and Hill’s coefficient (n_H_) equal to 5.0 ± 1.5. This IC_50_ does not differ significantly from the IC_50_ obtained earlier for the synthetic mixture of diastereomers in the same testing system (198 ± 19 vs. 227 ± 37 µM [16], *p* = 0.5054, *t*-test). Thus, optically pure sevanol shows effective inhibition of heterologously expressed rASIC1a with a potency similar to the synthetic diastereomer mixture.

### 2.2. Sevanol Effectively Inhibits Endogenous ASIC1a Currents in Neuronal-like Cells

Human neuroblastoma SH-SY5Y differentiated by all-trans-retinoic acid (RA) is broadly used as a model of dopaminergic neurons for the study of neuropathological conditions [18]. Recently, it was shown that RA-treated neuroblastoma abundantly expresses homomeric ASIC1a channels [19], an isoform primary for the CNS, so we further determined the inhibitory effect of sevanol on endogenous channels in this cell line using whole-cell patch–clamp electrophysiology. Sevanol inhibited pH 6.0-induced currents in a manner similar to that of heterologously expressed rASIC1a (Figure 2D). By fitting the dose–response data with Hill’s equation, the IC_50_ on hASIC1a was calculated as 222 ± 12 µM and n_H_ as 3.67 ± 0.29 (Figure 2C). This IC_50_ does not differ significantly from the value for rASIC1a (222 ± 12 vs. 198 ± 19 µM, *p* = 0.3167, *t*-test). Thus, sevanol effectively inhibits both rat and human ASIC1a with equal potency.

### 2.3. Currents of α1β2γ2 GABA_A_Receptor Are Positively Modulated by Sevanol

GABA_A_R, with the subunit composition α1β2γ2, take the largest share of the population of synaptic neuronal receptors [20]. Therefore, we tested sevanol against mouse α1β2γ2 GABA_A_ receptors heterologously expressed in *Xenopus laevis* oocytes. The ionic currents were elicited by the rapid application of 5 µM GABA. In the presence of sub- and low micromolar concentrations of sevanol, GABA-induced currents were significantly larger than control currents, indicating a positive modulation of the receptor by the compound (Figure 2F). No receptor activation by the application of sevanol alone was detected. The fitting of the dose–response data to Hill’s equation gave the concentration of half-maximal response (EC_50_) equal to 0.56 ± 0.04 µM, maximal efficacy (A_max_)—137.7 ± 0.6%, and n_H_—1.15 ± 0.09 (Figure 2E). Interestingly, the effect on GABA_A_R occurred at concentrations below 50 µM, at which sevanol displayed no inhibitory effect on ASIC1a. In addition, at 100 µM of sevanol (beginning of ASIC1a inhibition), GABA-induced currents did not significantly differ from control; i.e., any modulatory effects could not be detected (Figure 2E,F). This suggests that the mechanism of GABA_A_R modulation by sevanol is complex and depends on the dose.

### 2.4. Sevanol Alters Anxiety-Related Behavior in the Open Field and Passive Avoidance Tests

Since both ASIC1a and GABA_A_R are regulators of anxiety circuits, we hypothesized that sevanol could have a regulatory effect on fear-related states. Hippocampus plays a crucial role in regulating the memory of emotional behavior associated with anxious states [21,22]; therefore, we conducted a series of behavioral studies using an intracerebroventricular injection of sevanol (1 µL) at different concentrations (6 μM and 300 μM) into the hippocampus (Figure 3A).

The exploratory behavior of mice was assessed in the open-field test. With an infrared actimeter (Figure 3B), we observed that 6 μM of sevanol significantly reduced the distance traveled by the awake animals (Figure 3C, light green circles, *p* = 0.0180), indicating a decrease in global activity (Figure 3D, light green circles, *p* = 0.0296). Furthermore, 300 μM sevanol did not yield any significant effects compared to the control group, both in terms of distance traveled (Figure 3C, dark green circles, *p* = 0.1042) and global activity (Figure 3D, dark green circles, *p* = 0.0711). The number of rearing behaviors did not show significant differences when sevanol was administered at either 6 μM or 300 μM (Figure 3E, light and dark green circles, respectively, *p* = 0.1238, *p* = 0.8596). Additionally, we conducted a mechanical counting of fecal boluses (Figure 3F). Significant differences were observed when sevanol was administered at a concentration of 6 μM (Figure 3F, light green circles, *p* = 0.017), while the difference with the control was insignificant when sevanol was administered at a concentration of 300 μM (Figure 3F, dark green circles, *p* = 0.571). Also, we analyzed the time spent by the animals in the outer and inner zones of the open field. An increase in the time spent in the inner zone indicates a decreased level of anxiety [23], whereas an increase in the time spent in the outer zone (pressing against the wall or corners of the open field) corresponds to thigmotaxis [24], an indicator of anxiety-associated animal behavior. Mice administered with 6 μM of sevanol spent more time in the outer zone (Figure 3G, light green circles, *p* = 0.0296 vs. control) compared to mice administered with 300 μM of sevanol (Figure 3G, dark green circles, *p* = 0.4170), demonstrating higher levels of anxiety compared to the control group.

Next, we assessed the anxiety-related behavior of mice in the passive avoidance test using the shuttle box (Figure 4A). Passive avoidance for rodents is defined as a suppression of an innate preference forthe dark arm of the test box following exposure to an aversive stimulus (foot shock) associated with it. The developed delay before entering the dark arm denotes an avoidance of entering a punished area and is based on Pavlovian fear conditioning [25]. Animals administered with 6 μM of sevanol displayed a retention of the acquired fear, as reflected in the time spent transitioning between the arms. In contrast, mice administered with 300 μM of sevanol showed attenuated avoidance behavior, as compared to the control group (Figure 4B, light and dark green circles, respectively, *p* = 0.5, *p* = 0.0296), which implies that the compound disrupts neuronal circuits involved in fear learning. These results show that sevanol exerts a bidirectional regulation of fear-related functions in the hippocampus, which depends on the administered dose and a behavioral paradigm.

### 2.5. Behavioral Effects of Sevanol Are Paralleled by Alterations in Hippocampal θ Rhythm

To investigate the impact of sevanol on hippocampal rhythmic activity in animals, we recorded hippocampal electroencephalographic (EEG) brain activity during the open field test. The recorded frequencies corresponded to the θ range (5–10 Hz), which is associated with the animal’s movement. The control group of mice exhibited a signal with regular rhythmicity in the θ frequency range, indicating a synchronized θ rhythm [26]. In contrast, animals after sevanol injection of6 and 300 µM showed signals with more irregular rhythmicity in the θ frequency range, mixed with slower (5–7 Hz) and faster (8–10 Hz) θ components, indicating a desynchronized θ signal [27] (Figure 5A).

This observation was quantitatively analyzed by examining the spectral composition of the EEG using discrete Fourier transformation. The running speed of animals under normal conditions without pathologies is closely related to the amplitude and frequency of the θ oscillation [28,29]. In the analysis of EEG in mice administered with sevanol at a concentration of 6 μM, a decrease in signal amplitude in the θ frequency range was observed (Figure 5A,D, light green circles, *p* = 0.0468), and at certain moments, the signal completely attenuated, corresponding to a decrease in exploratory and locomotor activity of the animal in the open field (Figure 3B–D, light green circles). Mice administered with sevanol at a concentration of 300 μM also showed a decrease in signal amplitude in the θ frequency range (Figure 5A,D, dark green circles, *p* = 0.0296), while their exploratory and locomotor activity in the open field maintained (Figure 3B–D, dark green circles). Using continuous frequency–time spectrograms and Fourier spectrograms, we determined that mice treated with 6 μM of sevanol exhibited a shift of the θ frequency towards the lower range (Figure 5B,C, light green circles, *p* = 0.0156). In contrast, mice treated with 300 μM of sevanol showed either no shift or a change towards the higher θ frequency range (Figure 5B,C, dark green circles, *p* = 0.5).

Also, we conducted additional EEG recordings using GABA. In mice administered with 30 μM of GABA, we observed a decrease in signal amplitude (Figure 5E–G, dark green circles, *p* = 0.0404) and a reduction in frequency within the low-frequency θ range (Figure 5H, dark green circles, *p* = 0.0404), the effects similar to those observed for the group challenged with 6 μM of sevanol.

## 3. Discussion

In this study, we showed that sevanol, an inhibitor of ASIC channels, additionally positively modulates GABA_A_R. Both neuronal targets are profound for the functioning of CNS and are implicated in the regulation of fear and anxiety. The effect on α1β2γ2 GABA_A_R was pronounced in sub- and low-micromolar concentrations and was absent at 100 µM, unlike the effect on ASIC1a, which was detectable at this concentration. This led us to an assumption that sevanol could be used as a molecular tool to distinguish the roles of the two molecular targets present in the hippocampus in anxiety-related behavioral models upon the challenge with a low or high dose of the compound.

Positive allosteric modulators of GABA_A_R are able to demonstrate sedative, anxiolytic, or anticonvulsant effects, tuning inhibitory synaptic signaling throughout the central nervous system [30,31]. An openfield test is often used for the evaluation of the anxiolytic activity of compounds; however, this test produces a great variability in size and even direction of effects of most prescribed anxiolytics, including benzodiazepines [32]. In the open field test, 6 µM of sevanol, a dose active on GABA_A_R and inactive on ASICs, produced a significant sedative effect (Figure 3), decreasing global activity and traveled distance, together with anxiety-associated animal behavior (increased time in the outer zone and the number of fecal boluses). Reduced locomotor activity of mice and elevated thigmotaxis was paralleled by a desynchronized θ rhythm of the hippocampus with lower amplitudes and a shift to a lower-frequency range, which corresponds well to the recordings obtained with the administration of GABA and confirms that the low dose of sevanol primarily stimulates GABAergic neurotransmission.

At a dose of 300 µM, whichis inactive to α1β2γ2 GABA_A_R but inhibits ASICs, we did not observe any behavioral changes in the open field, although the shift to lower amplitudes in EEG was detected. Thus, although the inhibition of ASICs definitely interferes with neurotransmission, apparently, this is not manifested in the locomotor activity of mice. As reported previously, ASIC inhibitors or the loss of the ASIC1a gene may have no effect in models that measure baseline fear of open environments, but these compounds could be effective in fear models based on noxious and aversive stimuli [5,13,33,34]. Accordingly, we did not observe significant behavioral effects in the open field test after the treatment with the high dose of sevanol. However, we also should note that, in one study [7], knockout mice showed impaired center avoidance in the open field measured as breaks of center light beams.

In the passive avoidance test, we observed a significant suppression of the avoidance behavior in the group challenged with the high dose of sevanol (Figure 4). ASIC1a channels play a significant role in fear learning and memory [33]. To our knowledge, there are no reports on testing ASIC inhibitors in a passive avoidance model as used in this work; however, some behavioral effects could be expected. In a four-plate test, a somehow similar model of fear conditioning in which the exploratory behavior is suppressed by the delivery of a foot shock, an acute treatment of mice with ASIC1a inhibitors (PcTX-1 and A-317567) produced a significant increase in punished crossings compared with a control group [13]. As suggested previously, ASIC1a can play a role in the processing and coping responses when the animal is exposed to stressful and aversive stimuli [13]. We hypothesize that sevanol produces an amnesia of learned fear via the inhibition of ASICs in distinct hippocampal circuits, which is manifested in the suppression of the passive avoidance of a foot shock.

The discovery of an additional high-affinity target for sevanol could explain previous data on non-linear dose–responses when efficacy in low doses (0.01 mg/kg) was equal or superior to the effects of high doses (1–2.5 mg/kg) in a CFA-induced thermal hyperalgesia test and an acetic acid-induced writhing test [35]. Sevanol has the same effect on human ASIC1a as on rat channels (Figure 2); therefore, results of in vivo experiments corresponding to the inhibition of ASIC1a could possibly be translated to humans. Nevertheless, anxiolytic potential and the variety of effects on humans following treatment by different doses of sevanol via different routes of administration should be unambiguously evaluated only in clinical studies.

## 4. Materials and Methods

### 4.1. Synthesis and Purification of Sevanol

Sevanol/isosevanol was produced by chemical synthesis, as described in [17]. The preparation was a mixture of two diastereomers in a ratio of 1:3 (isosevanol:sevanol). The purification was performedby high-performance liquid chromatography (HPLC) using a reversed-phase column Luna C_18_(2) (250 × 10 mm) (Phenomenex, Torrance, CA, USA) with isocratic elution by 10% acetonitrile with 0.1% trifluoroacetic acid (TFA) or in a slow linear gradient of acetonitrile concentration with 0.1% TFA.

### 4.2. Cell Culture

Human neuroblastoma SH-SY5Y (CLS Cat# 300154/p822_SH-SY5Y, RRID:CVCL_0019) was obtained from ATCC (Manassas, VA, USA). Cells with a neuronal-like phenotype were cultured as described in [19]. In brief, SH-SY5Y cells were grown in Basic Growth Media (DMEM/F12 (1:1) (ThermoFisher Scientific, Waltham, MA, USA) with 2 mM L-glutamine, 10% fetal bovine serum (HyClone, Logan, UT, USA), 100 U/mL penicillin, 100 µg/mL streptomycin at 37 °C/5% CO_2_. For cell differentiation, on day 0, cells were seeded in Basic Growth Media to the confluence of 40–50%; on day 1, the medium was changed to the differentiation medium (DMEM/F12 (1:1)) with 2 mM L-glutamine, 1% fetal bovine serum, 100 U/mL penicillin, 100 µg/mL streptomycin, 10 µM all-trans-retinoic acid (Sigma-Aldrich, St. Louis, MO, USA), and cells were protected from light. Patch–clamp experiments were performed on days 6–8.

### 4.3. Whole-Cell Patch–Clamp Recordings

Recordings of whole-cell currents were producedwith an EPC-800 amplifier (HEKA Elektronik, Lambrecht, Germany), an InstruTECH LIH 8 + 8 data acquisition system (HEKA), and PatchMaster v2x90.5 software (HEKA). Patch pipettes were pulled from borosilicate glass with a P-1000 flaming/brown micropipette puller and had a resistance rangingfrom 5 to 8 MΩ when filled with the pipette solution containing (in mM) 110 K-gluconate, 20 KCl, 10 HEPES, 5 EGTA, pH 7.2 adjusted with 1 M KOH. The extracellular solution contained (in mM) 140 NaCl, 5 KCl, 2 CaCl_2_, 1 MgCl_2_, 15 D-glucose, 5 HEPES, pH 7.4 adjusted with 1 M NaOH. To buffer the activating solution (pH 6.0), HEPES in the extracellular solution was substituted for 10 mM MES. Ligands were applied after achieving a stable response. The holding membrane potential was −90 mV. Extracellular solutions were applied with a perfusion system controlled by an SF-77B Perfusion Fast Step (Warner instruments, Holliston, MA, USA).

### 4.4. Xenopus Laevis Oocytes

This study was performed in strict accordance with the World Health Organization’s International Guiding Principles for Biomedical Research Involving Animals. The protocol was approved by the Institutional Policy on the Use of Laboratory Animals of the Shemyakin-Ovchinnikov Institute of Bioorganic Chemistry RAS (Protocol Number: 267/2018; date of approval: 28 February 2019). All procedures were performed in agreement with the guidelines of ARRIVE (Animal Research: Reporting of In Vivo Experiments) and the “European convention for the protection of vertebrate animals used for experimental and other scientific purposes” (Strasbourg, 18.III.1986). Oocytes were harvested from female frogs anesthetizedwith tricaine methane sulfonate (MS222) (0.17% solution), and the surgery was performed in an ice bath to avoid heavy bleeding. Defolliculated stage IV and V cells were injected with 2.5 ng cRNA, synthesized from pCI plasmids containing genes of rat ASIC1a or murine α1, β2, and γ2 GABA_A_ subunits, using the Nanoliter 2000 microinjection system (World Precision Instruments, Sarasota, FL, USA). The injected oocytes were kept for 2–3 days at 17 °C, and then for up to 5 days at 15 °C in a ND96 medium containing (in mM) 96 NaCl, 2 KCl, 1.8 CaCl_2_, 1 MgCl_2_, and 5 HEPES, titrated to pH 7.4 with 1 M NaOH, and supplemented with gentamycin (50 µg/mL).

### 4.5. Two-Electrode Voltage-Clamp Recordings

Microelectrodes pulled from borosilicate glass with a P-1000 flaming/brown micropipette puller (Sutter Instrument, Novato, CA, USA) were filled with 3 M KCl. For ASIC channels, recordings were performed at a holding potential of −50 mV using a GeneClamp 500 amplifier (Molecular Devices, San Jose, CA, USA). The conditioning bath solution was ND96 (pH 7.4). For solutions with pH 6.0, 5 mM HEPES was replaced with 10 mM MES. For GABA_A_ receptors, the membrane potential was clamped at −70 mV using an Axon amplifier (Molecular Devices, LLC., San Jose, CA, USA). The data were collected, digitized, and analyzed using AxoScope 9.2.1.9 software (Molecular Devices, LLC., San Jose, CA, USA). Applications of GABA were performed every 5 min. The tested compounds were applied 4 min before GABA application. Amplitudes of GABA-evoked currents were measured and normalized to control amplitudes of the solo GABA-evoked current response.

### 4.6. Mice

The protocol of experiments on mice was approved by the Institutional Policy on the Use of Laboratory Animals of the Shemyakin-Ovchinnikov Institute of Bioorganic Chemistry RAS (Protocol Number: 353/2023; date of approval: 3 July 2023). Five- to eight-week-old C57BL/6 male mice were used. Mice were kept in individual cages with food and water ad libitum and maintained on a 12:12-h light-dark cycle.

### 4.7. Surgical Procedure of Electrode Implantation and Cannula Placement

Implantation of the manufactured electrodes and cannulas into mice for chronic experiments was carried out according to previous studies [36]. Before the operation, the mouse was weighed and placed in the euthanasia chamber. Three to four percent isoflurane was applied over two to threemin, and prior to transferring to the stereotaxic frame, lidocaine (10 mL/kg, 0.5% subcutaneous (s.c.)) was injected, and the animal’s skull was prepared for the operation by shaving hair. The animal was then transferred to the stereotaxic frame with 1–2% isoflurane at a rate of 1 mL/min. The skin in the area of interest was treated with 45% ethanol and 5% iodine. Reflexes were checked by clamping the animal’s paw with forceps to confirm complete anesthesia. To prevent eye drying throughout the surgery, Vidisic eye gel (0.2%, GmbH) was applied. A 10 mm diameter incision was createdover the target brain region, and the skin was removed, exposing the skull bone. The bone was cleaned withphysiological saline after 0.2% hydrogen peroxide. Notches were createdon the skull bone using a drill (to create a better contact between the bone and dental cement). By using an animal stereotaxic atlas [37], the coordinates for electrode implantation and cannula placement were determined. Two stainless steel screws were mounted into the frontal bone, and one ground stainless steel screw was mounted into the parietal bone for the fixation of the whole structure with dental cement (Meliodent). Holes for electrode implantation were drilled into the skull bones in one of the hippocampi, and the cannulas were placed in the projection of both hippocampi. A bundle of electrodes was constructedfrom nickel–chromium wires (Science Products) soldered to connector headers (SLR1025 G, Fischer Elektronik, Lüdenscheid, Germany). Cannulas with a length of 6 mm were constructedfrom surgical needles (23GA) and internal stoppers from insulin needles (31GA). The coordinates used for electrode implantation were 1.7 mm posterior to the bregma, −2.0 mm lateral to the midline, and −1.9 mm below the dura mater. For cannulas, the coordinates were 2.1 mm posterior to the bregma and ±2.0 mm lateral to the midline. The entire surgical site was filled with dental cement (Meliodent), and the skin was glued with tissue adhesive (Sulfacrylate). After the surgery, ketoprofen (5 mg/kg s.c.) and saline were administered under the nape, depending on the animals (0.1–0.5 mL). For postoperative care and analgesia, ketoprofen was provided (5 mg/kg per o.s. or s.c. once daily). The experimentaldesign is shown in Figure 3A.

### 4.8. Drug Injection for Behavioral Tests

All drugs were kept as stock concentrations at −20 °C. Sevanol and γ-aminobutyric acid (GABA, Tocris Cookson, UK) were diluted to final concentrations in Ringer’s solution containing (in mM) 127 NaCl, 2.5 KCl, 1.25 NaH_2_PO_4,_ 1 MgCl_2_, 2 CaCl_2,_ 25 NaHCO_3_, 25 D-glucose. The drugs were injected in both injection sites, each at a volume of 1 μL, 1.5 h prior to the start of behavioral experiments. For the control group mice, Ringer’s solution was used. The animal was anesthetized with 5% isoflurane and then moved to a stereotaxic frame with a constant supply of 1–2% isoflurane by 1 mL/min. For the intrahippocampal injection, an UltraMicroPump III with Micro4 Controller system (World Precision Instruments, Friedberg, Germany) with a Hamilton syringe and a NanoFil (36GA) beveled needle (World Precision Instruments, Friedberg, Germany) was used. The needle was lowered through a guide cannula to a depth of 1.9 mm from the brain surface into the hippocampal region. The fluid delivery program was initiated at 1 nL/s with a volume of 500 µL. For better targeting, we waited 10 min after the injection before the needle was retracted.

### 4.9. Behavioral Tests

After the recovery period, the mice were subjected to behavioral tests such as open field and passive avoidance, as well as recording hippocampal rhythmic brain activity. Behavioral tests were conducted between 10 a.m. and 10 p.m. to carry out experiments with awake animals. The interval between the control and comparison group tests was 24 h. To reduce stress in the behavior of animals, prior to the experimental period, mice were given the opportunity to habituate to the experimenter’s hands for 2 days (handled 2–3 times a day) [38]. All behavioral experiments were conducted in a dark room with red lighting, as the animals were not sensitive to this light source and felt comfortable [39].

### 4.10. Open Field Test

The Actitrack v2.55 software (Panlab, S.L.U., Cornellà (Barcelona), Spain) was used to set up the infrared (IR) actimeter with the upper IR frame adjusted to the height of the mouse’s rearing. The experiment used a mouse rearing duration threshold of 2.5 s, the time required for the mouse to stand up on its hind legs. Mouse velocity thresholds were set to 5.5 cm/s for slow movement and 10 cm/s for fast movement. Two zones, central and peripheral, were set up for detecting the mouse’s movement, with 5 min allotted for the test. Both the control and comparison groups of mice were individually transferred from their home cages to the IR actimeter, given 5 min to explore the unfamiliar territory, and then returned to their home cages. The apparatus was wiped with 70% ethanol before transferring each subsequent mouse. This test was repeated throughout the experimental period with the administered drugs.

### 4.11. Passive Avoidance Test

The mice were sequentially placed in a shuttle box with both arms devoid of any conditioned and unconditioned stimuli and allowed to explore the chambers for 5 min before being returned to their home cages. Both arms were wiped with 70% ethanol. At this stage, no setting was required for the ShutAvoid software (Panlab SHUTTLE-8 software, S.L.U., Cornellà (Barcelona), Spain). After 1 h, a training session was conducted in which the animals were placed in the bright arm of the shuttle box, the door between the arms was lowered, and an unconditioned stimulus (foot shock) was administered in the dark arm. After 1 min, the door was raised, and the animal was given 2 min to transition between the arms before being returned to its home cage. Both arms were wiped with 70% ethanol. The ShutAvoid software was configured with the following settings: the time interval before the door was raised was 60 s, the time given to transition from the bright arm to the dark arm was 120 s, the delay before lowering the door after entry into the dark arm was 5 s, and the duration of the unconditioned stimulus was 2 s. In addition, if the mouse entered the dark arm after the experiment, a 60 s period was given. The power of the unconditioned stimulus was set to 0.5 mA.

After 24 h, a retention session was conducted for the control group animals, in which they were placed in the bright arm of the shuttle box, the door between the arms was lowered, and there was no unconditioned stimulus in the dark arm. After 30 s, the door was raised, and the animal was given 2 min to transition between the arms before being returned to its home cage. Both arms were wiped with 70% ethanol. The ShutAvoid software was configured with the following settings: the time interval before the door was raised was 30 s, the time given to transition from the bright arm to the dark arm was 120 s, and the delay before lowering the door after entry into the dark arm was 5 s. After 24 h, a retention session was conducted for the comparison group animals (6 μM sevanol), and the manipulations and settings were identical to those for the control group. After 24 h, a retention session was conducted for the comparison group animals (300 μM sevanol), and the manipulations and settings were identical to those for the control group.

### 4.12. In Vivo Electrophysiology

For recording hippocampal local field potentials (LFPs), the wireless in vivo system (MCS) was used in conjunction with Multi Channel Suite software (Multi Channel Systems MCS GmbH, Reutlingen, BW, Germany). The following settings were used in the Multi Channel Experimenter of Multi Channel Suite software: sample rate 5 kHz, electrical voltage ±2.5 V, recording time 300 s. Prior to LFP recordings, a wireless head stagewith 8 recording channels (MCS) was placed at the contact site of the animal’s head and left in the home cage until full recovery was achieved, typically 30–40 min, after the manipulations described in Section 3. The mouse was then transferred to an “open field” IR actimeter, where the hippocampal rhythmic activity of the awake animal was recorded during the study period. Afterward, the mouse was returned to the home cage, and the wireless headstage was removed. Prior to starting subsequent animals, the IR actimeter and wireless headstage were wiped with 70% ethanol. All raw signals were stored for offline analyses. The raw LFP recordings were saved and converted to the ASCII format using the native Multi Channel DataManager of Multi Channel Suite software.

### 4.13. Construction of Continuous Time–Frequency Spectrogram for θ Rhythm

The data processing was performed using MATLAB 2013b (The Mathworks Inc., Natick, MA, USA) and Origin 8 (OriginLab Corp., Northampton, MA, USA) software. First, each raw signal segment was divided into 60-s epochs and then subjected to band-pass filtering within the 3 to 15 Hz range, preserving the low- and high-frequency theta range. The power spectra were calculated with a resolution of 0.25 Hz using the Welch method with a Hamming window (pwelch.m function from the Signal Processing Toolbox). The area under the curve (integral) for power spectra was calculated using the cumtrapz.m function. The power spectra were normalized by dividing them by the calculated integrals so that the total area under the spectrum curve was equal to 1. The resulting spectra were exported to Origin 8 software for detailed visualization. The continuous time–frequency Morlet wavelet transform was constructed using the cwt.m function [40,41]. The data matrix from MATLAB was exported to Origin 8 software for detailed visualization (Figure 5A,E). A package of scripts based on these functions was developed to simplify all the manipulations described above, and it is available upon request.

### 4.14. Histology

The mice were anesthetized with isoflurane before being sacrificed, and their brains were collected by dissection. The extracted material was fixed in ethanol in three stages with a 24-h interval between each stage [42]. The material was then sliced on a vibrating microtome (Leica, Wetzlar, Germany) in 60% ethanol at a thickness of 100 µm. Methylene blue dye (HiMedia Laboratories, Thane, India) was used to determine electrode location [42,43], and the sections were mounted on slides with axylene-based medium (DiaPath). After 24 h, the site of electrode traces was detected using an optical microscope (Leica) (Figure 5I).

### 4.15. Data Analysis

In vitro dose–response data were analyzed using PatchMaster, OriginPro 8.6 (OriginLab, Northampton, MA, USA), and GraphPad Prism 7.00 (GraphPad Software, San Diego, CA, USA). Dose–response data were fitted with a Hill equation Ix = I_0_/[1 + ([x_0_]/[x])^n_H_], where I_x_ is the ionic current amplitude at a given concentration of ligand [x], I_0_ is the ionic current in the absence of ligand, x_0_ is the concentration at which a ligand exerts half of its maximal effect, and n_H_ is the Hill coefficient. The EEG data were acquired using a multi-channel system interface (MCS) and analyzed using Multi Channel Analyzer software (MCS) and MATLAB 2013b (The Mathworks Inc., Natick, MA, USA). Behavioral data (mice tracking) were obtained using the Infrared (IR) actimeter interface (Panlab, Barcelona, Spain) and analyzed using Actitrack software (Panlab). Passive avoidance data were obtained using the Shuttle box interface (Panlab) with a shock generator (Panlab) and analyzed using Shutavoid software (Panlab). All data are presented as the mean ± SEM. For in vitro data, the differences between groups were tested with Mann–Whitney test (GraphPad Prism 9.5.1). For in vivo data, with paired sample Wilcoxon signed-rank test and Mann–Whitney test (Origin 8). Differences with *p* < 0.05 were considered significant.

## 5. Conclusions

We showed that sevanol positively modulates the currents of α1β2γ2 GAB_A_A receptors in a sub- and low-micromolar range, in addition to the known antagonism of ASIC1a and ASIC3 channels. Intriguingly, intervals of efficacious concentrations for GABA_A_R and ASICs do not overlap. This fact allowed us to compare the effect of sevanol on CNS as a GABA_A_R-positive modulator and as an antagonist of ASIC1a. The low dose of sevanol (i.c.v.) induced a sedative effect in the open field test, and changes in rhythmic brain activity according to the recordings of hippocampal electroencephalographic rhythm were similar to GABA. Administration of a dose capable of inhibiting ASIC1a had no effect in the open field but significantly attenuated avoidance behavior in the passive avoidance test. Pharmacological effects on both targets are considered advantageous, but this property of sevanol should be taken into account in drug development and basic research.

## Figures and Tables

**Figure 1 ijms-24-13148-f001:**
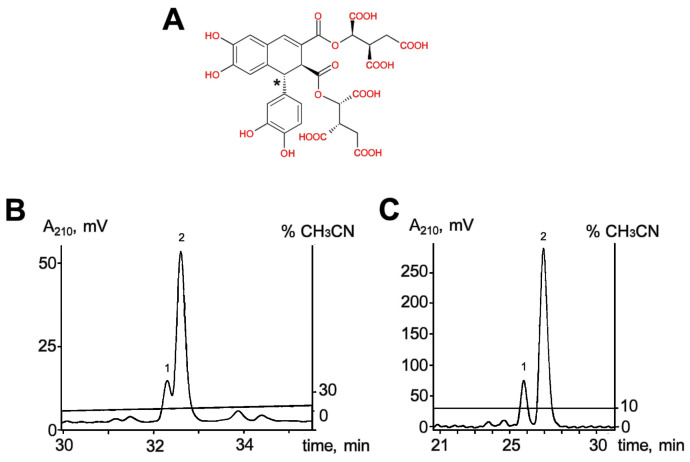
(**A**) Chemical structure of sevanol. The asterisk designates the chiral center in which sevanol and isosevanol differ in configuration. (**B**,**C**) Separation of diastereomers, 1—isosevanol, 2—sevanol, on a reversed-phase column Luna C_18_(2) (250 × 10 mm) by elution in a linear gradient of acetonitrile with 0.1% TFA (50 µg of the mixture, panel B) and by isocratic elution in 10% acetonitrile with 0.1% TFA (300 µg of the mixture, panel C).

**Figure 2 ijms-24-13148-f002:**
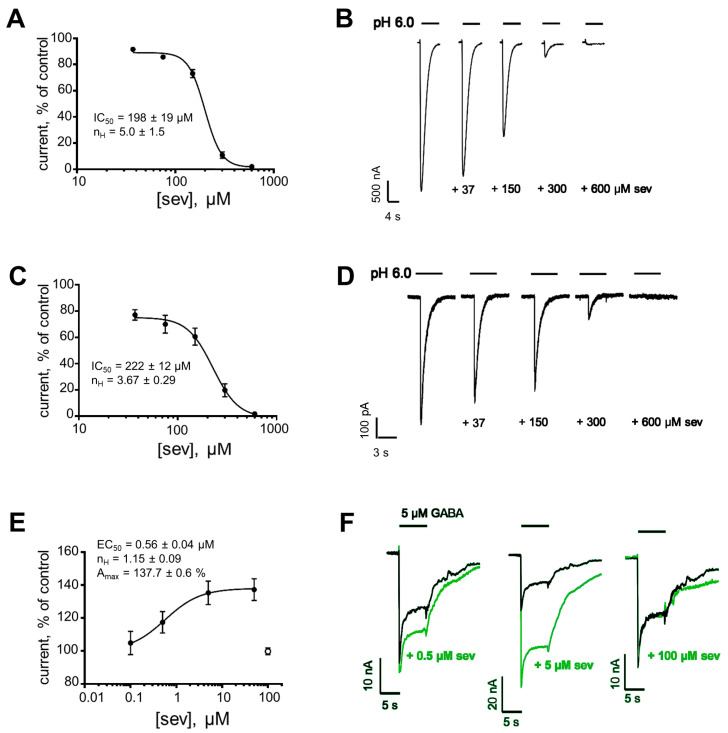
In vitro activity of optically pure sevanol (sev). (**A**) Dose–response curve on rASIC1a currents measured on *Xenopus* oocytes, IC_50_ = 198 ± 19 µM, n_H_ = 5.0 ± 1.5. Each point represents data from 6–7 cells. (**B**) Representative traces of rASIC1a currents induced by pH 6.0 from conditioning at pH 7.4, with and without the pre-application of sev. (**C**) Dose–response curve on hASIC1a currents in RA-treated SH SY5Y cells, IC_50_ = 222 ± 12 µM, n_H_ = 3.67 ± 0.29. Each point represents data from 4–5 cells. (**D**) Representative traces of hASIC1a currents induced by pH 6.0 from conditioning at pH 7.4, with and without the pre-application of sev. (**E**) Dose–response curve on mouse α1β2γ2 GABA_A_R currents, EC_50_ = 0.56 ± 0.04 µM, n_H_ = 1.15 ± 0.09, A_max_ = 137.7 ± 0.6%. White point was excluded from the fitting. Each point represents data from 4–11 cells. (**F**) Representative traces of mouse α1β2γ2 GABA_A_R currents from a single cell induced by 5 µM GABA in the presence of sev. The data are presented as the mean ± SEM.

**Figure 3 ijms-24-13148-f003:**
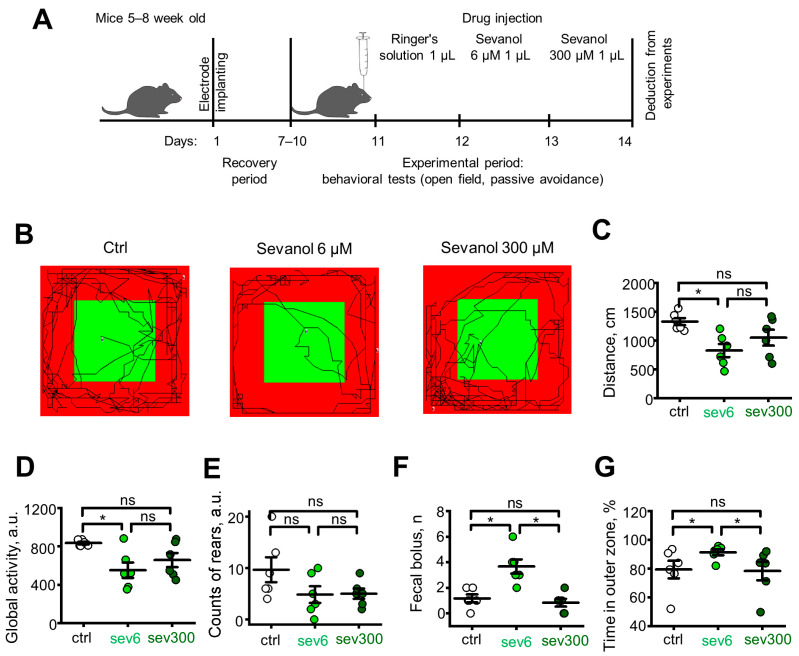
Effect of sevanol in the open field test. (**A**) Experimental design. (**B**) Representative tracks of mice activity for 5 min. (**C**) Total distance traveled by animals. (**D**) Global activity indicator. (**E**) Counts of rodent racks. (**F**) Number of fecal boluses. (**G**) The time spent in the outer zone of the open field. ctrl—control group (mice administered with vehicle), sev6—mice administered with 6 µM sevanol, sev300—mice administered with 300 µM of sevanol. The data are presented as the mean ± SEM. ns *p* > 0.05; * *p* < 0.05; paired sample Wilcoxon signed-rank test, n = 6 for each group.

**Figure 4 ijms-24-13148-f004:**
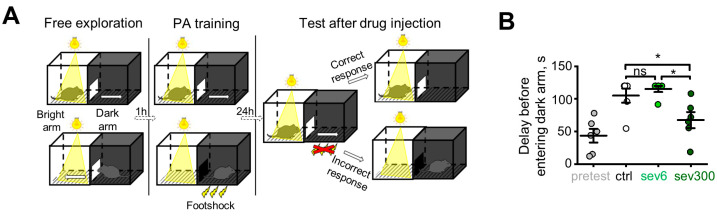
Effect of sevanol in the passive avoidance test. (**A**) Schematic depiction of passive avoidance experiment. (**B**) Time delay to enter the dark arm. Ctrl—control group (mice administered with vehicle), sev6—mice administered with 6 µM of sevanol, sev300—mice administered with 300 µM of sevanol. The data are presented as the mean ± SEM. ns *p* > 0.05; **p*< 0.05; paired sample Wilcoxon signed-rank test, n = 6 for each group.

**Figure 5 ijms-24-13148-f005:**
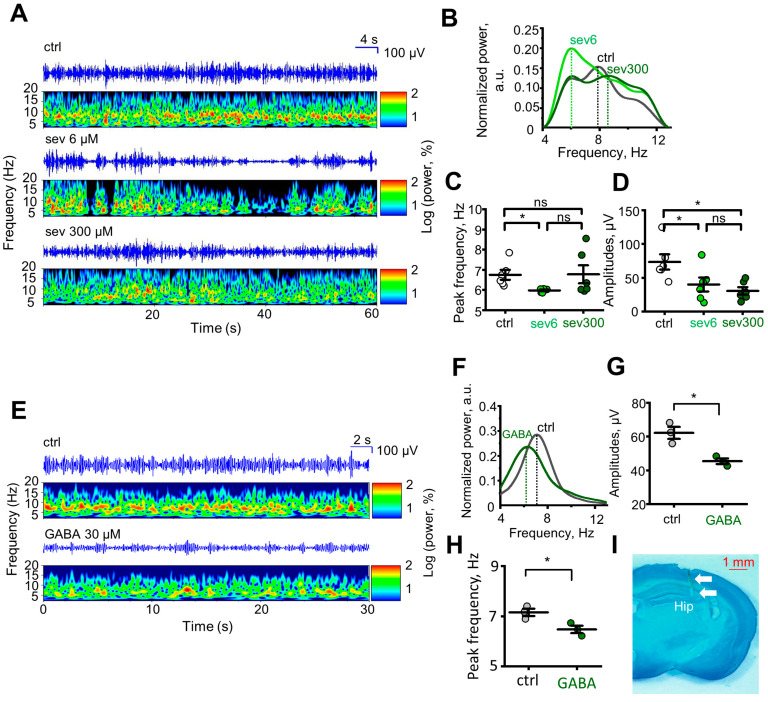
Effect of sevanol and GABA on the rhythmic activity of the brain. (**A**) Continuous time–frequency spectrograms of the local field potentials (LFP) during 60 s of the open field exploration by the control group and groups administered with 6 and 300 μM of sevanol. (**B**) Representative traces of frequency shift in θ range after sevanol administration. (**C**) Peak frequency θ range after sevanol administration. (**D**) Amplitude changes after sevanol administration. (**E**) Continuous time–frequency spectrograms of LFP during 30 s of the open field exploration by the control group and group administered with 30 μM of GABA. (**F**) Representative traces of frequency shift in θ range after GABA administration. (**G**,**H**) Amplitude change (**G**) and peak frequency θ range (**H**) after GABA administration. (**I**) Example tissue slice, showing the site of electrode traces (white arrows). Ctr—control group (mice administered with vehicle), sev6—mice administered with 6 µM sevanol, sev300—mice administered with 300 µM of sevanol. The data are presented as the mean ± SEM. ns *p* > 0.05; * *p*< 0.05; paired sample Wilcoxon signed-rank test (panels c, d), n = 6 for each group; Mann–Whitney test (panels g, h), n = 3 for each group.

## Data Availability

The data that support the findings of this study are available from the corresponding author upon reasonable request.

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
