# Peer review of "Dual Modulator of ASIC Channels and GABA_A_ Receptors from Thyme Alters Fear-Related Hippocampal Activity"

_ijms, 2023, doi:10.3390/ijms241713148_

Round 1
Reviewer 1 Report
This study led by Kalinovskii et al. examined the dual modulation of ASIC and GABAA receptor by sevanol. The authors showed that sevanol inhibits ASIC1a at a higher concentration, whereas it positively modulates GABAA receptor at a relatively lower concentration. Behaviorally, local microinjection of a lower dose of sevanol into the hippocampus results in reduced locomotor activity of mice and elevated thigmotaxis. Electrophysiologically, sevanol triggered a desynchronized θ rhythm of hippocampal neurons with lower amplitudes and a shift to lower-frequency range, probably by stimulating GABAergic neurotransmission. On the other hand, sevanol at a higher dose markedly suppressed the passive avoidance behavior, possibility via inhibition of ASICs in the hippocampal circuits. I have several questions required to be addressed to improve the overall quality of the paper.
1. It is of great interest to test whether sevanol affects other ASIC isoforms, such as ASIC2a, ASIC2b, ASIC3 and the various heteromers. Also, please check the effect of sevanol on ASIC1a currents at different pH values.
2. Fig. 3E. The tile of Y axis should be “Counts of rears” but not “Counts of racks”.
3. Fig. 3. Does sevanol-induced hypolocomotion rely on its action on GABAA receptors? If this is true, direct administration of GABA should recapitulate the behavioral effects of sevanol.
4. Fig. 4. The authors speculated that sevanol (300 μM) impaired the passive avoidance behavior through inhibiting the activation of ASIC1a. Then one necessary experiment involves testing this behavioral effect with the ASIC1a KO mice. It is expected that sevanol would not affect the passive avoidance in the ASIC1a KO mice? Or else, the ASIC1a inhibitors would mimic the effect of sevanol to produces an amnesia of learned fear. These possibilities should be tested to further strengthen the assertion made by the authors.
5. Roles of ASIC1a in fear memory have been extensively studied in the amygdala. Comparatively, much less information is available on the function of ASIC1a in the dorsal hippocampus in mediating or modulating fear behavior. So please explain or discuss the rationale why this study focuses on the behavioral action of sevanol in the hippocampus but not try amygdala.
Reviewer 2 Report
This work reports on the electrophysiological characterization of the synthetic form of a natural lignan, sevanol, from thyme (Thymus armeniacus) on acid-sensing ion channels (ASIC) and ionotropic GABA receptors (GABAAR). The compound was synthesized and, when tested, it inhibited homomeric rat ASIC1a (expressed in oocytes of Xenopus laevis and in human neuroblastoma cells SH-SY5Y differentiated by all-trans retinoic acid) at high microM and positively modulated mouse α1β2γ2 GABAAR (expressed in Xenopus laevis oocytes) at sub- and low microM.
The authors also tested the compound in two anxiety-related behavior models, in mouse, by delivering it into hippocampus and employed accompanying electroencephalographic recordings. In the open field test, sevanol (6 μM) reduced both locomotor and θ-rhythmic activity like GABA, which suggests a primary action on the GABAergic system. In the passive avoidance test, sevanol (300 μM,) evidently suppressed this behavior, which implies alterations in the conditioned fear memory.
The authors conclude that the effects of sevanol might be associated to different mechanisms that concern, respectively, the GABAAR and ASIC1a channels.
Thus, this work contributes to the characterization of the pharmacological profile of sevanol as a candidate for drug advance and supports the function of ASIC in hippocampal functions associated to fear. However, the authors state that clinical studies are necessary to evaluate the anxiolytic potential and other effects on humans.
MINOR POINTS:
- A few punctuation marks should be added or changed.
- A few words should be added, omitted, or changed.
- A few letters should be added or changed, including unnecessary uppercase letters in the titles of some articles.
- A few details on suppliers should be added or changed.
- A few words should be changed to their abbreviations.
(please, see the uploaded file)

Author Response
We thank the Reviewer for their interest and valuable corrections.
- A few punctuation marks should be added or changed.
- A few words should be added, omitted, or changed.
- A few letters should be added or changed, including unnecessary uppercase letters in the titles of some articles.
- A few details on suppliers should be added or changed.
- A few words should be changed to their abbreviations.
(please, see the uploaded file)
We accept all the corrections made by the Reviewer with few exceptions.
- Line 388. The mention of ggplot2 packacke analysis was deleted because this analysis did not end up in the final version of the manuscript. The traces presented in the manuscript were analyzed with GraphPad Prism and Origin (mentioned in section 4.15).
- Line 507. The term “epochs” is more commonly used in this context.
- The reference list was formed automatically using Mendeley so it may need manual corrections. We would like to make final corrections after the proofreading by the processing editor according to the journal’s requirements.
Reviewer 3 Report
This manuscript presents a study of the effect of a compound, sevanol, on the behavior of mice in fear inducing situations. The work appears to be competently done; in cases where the effects are significant, they tend to be significant at the p<.05 level. In other words, the effects are small, and marginally significant. The work has two stages: 1) in Fig. 2, the effect of the compound on the two channels is presented, as standard electrophysiology 2) in the main section of the paper, the effects of the compound on mouse behavior are shown.
By assuming that the compound acts through its effect on ASIC and GABA channels, and assuming there are no other effects, a correlation between the observed behavioral effects and the effects on the two classes of ion channels is inferred. While supported by the electrophysiological data (Fig 2) the reason why the lower concentration of sevanol affects GABA channels, but the higher concentration was not was not discussed. This appears to be a somewhat odd effect, and is worth some discussion. Is it possible that there are two effects that work in opposite directions? I am not aware of other cases in which a higher concentration has less effect than a lower concentration. This may not be entirely necessary in a paper that does not consider structural effects on the channels themselves, but should at least be remarked on. Considering that the compound has six carboxyl groups, is it possible that the extent of ionization is involved? At least a titration curve of sevanol should be included, with a brief discussion.
Other than this, the paper may be of some interest, and can be published as is.
